# Radiometric Normalization Using a Pseudo−Invariant Polygon Features−Based Algorithm with Contemporaneous Sentinel−2A and Landsat−8 OLI Imagery

**Lei Chen *, Ying Ma, Yi Lian, Hu Zhang, Yanmiao Yu and Yanzhen Lin**

School of Geographic and Environmental Sciences, Tianjin Normal University, Tianjin 300387, China
* Correspondence: chenleii0106@126.com

**Abstract:** As sensor parameters and atmospheric conditions create uncertainties for at−sensor radiation detection, radiometric consistency among satellite images is difficult to achieve. Relative radiometric normalization is a method that can improve multi−image consistency with accurate pseudo−invariant features (PIFs), especially over large areas or long time series satellite images. Although there are algorithms that manually or automatically select PIFs, the spatial mismatch of satellite images can affect PIF extraction, particularly with artificial pixels. To alleviate this problem, we proposed to use Landsat−8 OLI as the reference image and Sentinel−2A as the subject image, to apply pseudo−invariant features−based algorithms with polygon features through the single−band and multiple−band regression. Compared to pseudo−invariant point features, hyperspectral library, and histogram matching approaches, the results demonstrate the superiority of the proposed algorithms with correlation coefficients of 0.9948 and 0.9945, and an RMSE of 0.0097 and 0.0095 with multiple− and single−band regression, respectively. We also found more accurate linear fitting and better shape matching through band scattering and reflectance frequency analysis. The proposed algorithms are a significant improvement in radiometric normalization, within artificial pixels, achieving spectral signature consistency.

**Keywords:** pseudo−invariant polygon features; single−band regression; multiple−band regression; contemporaneous satellite images

## 1. Introduction

Remote sensing has proven itself to be essential in land change detection [1], crop growth monitoring [2], and forest succession monitoring [3]. Different kinds of satellite image data, including Landsat, MODIS, Sentinel, GF, and ZY, among others, are achieving great success. Building on this success, by integrating different types of satellite data, has the potential to solve ecological, natural resource, and human development challenges through large coverage remote sensing images, capitalizing on our ability to process large datasets [4–6]. However, among platforms, images differ due to variation in temporal, spatial, radiometric, and observation angles. Solving image data inconsistency is the most important issue that will allow the application of multiple remote sensing images. Radiometric value is essential in remote sensing, so radiometric normalization, which is a preprocessing procedure to eliminate radiometric differences among images, has been studied by several researchers with the development of many radiometric normalization algorithms [7–10].

Radiometric normalization algorithms can be divided into two main categories based on the transformation of grayscale values to physical signals: absolute radiometric normalization requiring accurate sensor calibration parameters and atmospheric properties [11,12], and relative radiometric normalization (RRN), which is an image−based approach using one image as a reference to normalize another image through its radiometric characteristics [13–15]. Due to parameter differences among sensors and the difficulty in collecting

atmospheric parameters, the relative radiometric normalization method is generally used for image normalization that does not require extra parameters [16,17]. Hence, in this process, the subject image tends to need similar radiometric conditions as the reference image. Nevertheless, images from different satellite sensors have inherent bandwidth and spectral response differences (among others) that limits the practical application of absolute radiometric normalization but facilitates relative radiometric normalization algorithms in many applications.

Among these relative radiometric normalization techniques, linear regression using pseudo−invariant features (PIFs) has been promoted as an empirical and practical method to normalize radiometric distortion caused by non−land surface−related factors [18,19]. This approach assumes that reference and subject images have a consistent response and adjusts the subject image's radiometric properties with paired pixels from the two images [20,21]. The key to preprocessing is to extract accurate PIFs, allowing image data collected at different times or by different sensors to be normalized. Comparing seven relative radiometric normalization methods with Landsat MSS images at different times, Sadeghi, Ebadi [19] and Syariz, Lin [22] have shown that accurately extracted PIFs can improve image radiometric consistency for the same sensor type at different times. Accurate PIFs are essential to compare sensor images. Rahman, Hay [7], Razzak, Mateo−García [16], Yan, Yang [17], and Padró, Pons [23] normalized images from different sensor types (e.g., Landsat, Sentinel−2, Gaofen, and super−resolution images) using a PIF−based method and reported coherent image correction among varying time series and sensor types. The biggest difference between these studies is the processing of manually or automatically PIFs selection.

Automatically extracting PIFs makes the radiometric correction processing more efficient and has been proposed by several researchers [24–28]. In these studies, PIFs are extracted automatically and sorted into a radiometric control set, radiometric control points, no−change set, and unchanged pixels [29–32]. However, manual PIFs extraction is more accurate in some applications, especially when ground types are complicated. Manual extraction allows pairwise pixels in different ground types to be selected, decreasing the matching error for paired pixels [7,15]. Although spatial mismatch problems in RRN are well−addressed in [18,33,34], three issues remain: (1) the spectra of paired pixels may differ due to the pixel−level spatial mismatch, especially for pairwise images with different sensing angles that create spatial distortion; (2) a regression analysis in which one variable does not consider the influence of adjacent bands, and; (3) applications are not focused on crop−growing areas and extracted PIFs are untrained to different types of crop or ground materials. To address these issues, we propose new relative radiometric normalization methods by replacing invariant points or pixels with invariant polygons or surfaces to apply to a crop−growing area that has been surveyed on the ground. We show decreases in the influence of pixel−level spatial mismatch and demonstrate the effect of adjacent bands using multiple variable regression. The proposed algorithms show superiority in crop area radiometric normalization and contribute to the improvement in normalization accuracy, particularly in human−impacted areas of the Earth's surface.

## 2. Study Area and Data

To test the performance invariant polygon and adjacent bands, we evaluated Landsat−8 OLI as the reference image and the Sentinel−2A as a distortion image. The study area is located at Baiquan, in Heilongjiang Province, China. Images were from September 2019 (Figure 1). In this area, crops, including rice, corn, and soybean, are close to harvest, resulting in mixed pixels with soil properties, contributing to the generalizability of our results. Field surveys identified five types of ground material: water, grass, trees, pure soil, and artificial. A regression relationship of minimum and maximum reflectance [20] proved to be the most accurate to predict land cover. Through field experiments, we established the relationship between two images based on the reflectance of different land cover types, including minimum through maximum reflectance values.

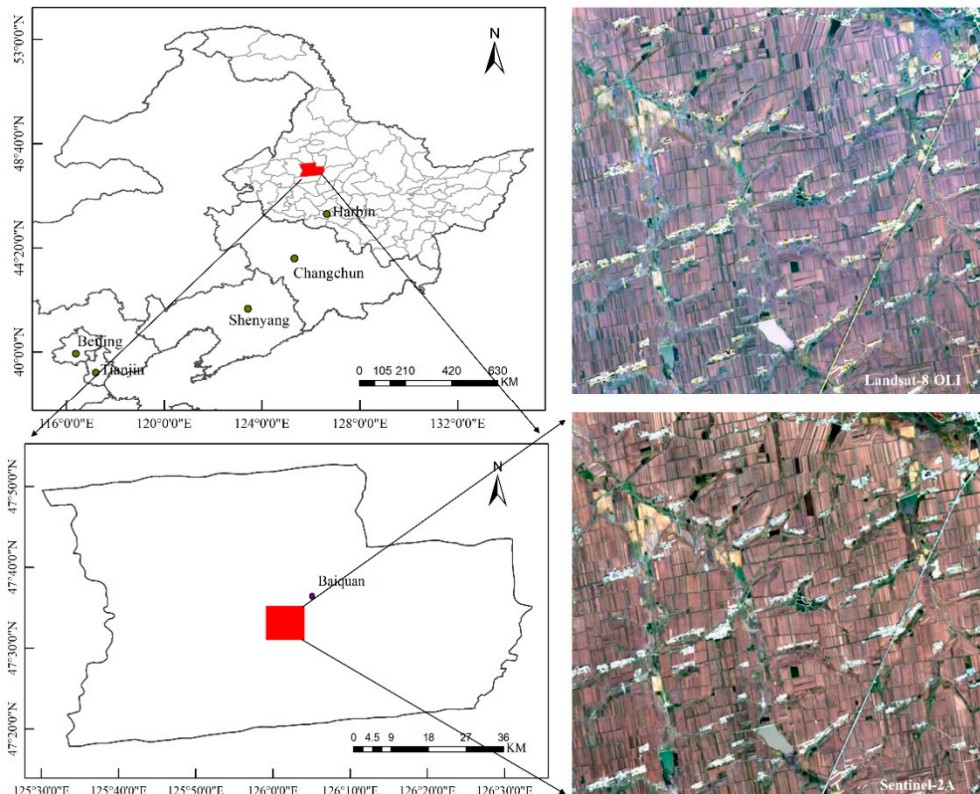

**Figure 1.** Scope and true color images of the study area.

To focus on radiometric normalization, we geometrically registered two images covering the same spatial range. To match tiles, we upscaled Sentinel−2A images using the nearest neighbor method to obtain the same spatial resolution as the matching Landsat−8 OLI image. Data were downloaded from the European Space Agency (https://sentinels.copernicus.eu/, accessed on 26 September 2019) and the United States Geological Survey (https://earthexplorer.usgs.gov/, accessed on 22 September 2019), both with absolute radiometric calibration. Here, we regard the Landsat−8 OLI image as a reference and Sentinel−2A as the subject. Six data bands were chosen for analysis (Table 1). Due to higher spatial resolution, the 0.785–0.900μm band was selected from Sentinel−2A imagery.

**Table 1.** The selected bands of image data.

| Image<br>Band | Landsat 8−OLI (Resolution) | Sentinel−2A (Resolution) |
|---|---|---|
| 1 (Blue) | 0.450–0.510 μm (30 m) | 0.458–0.523 μm (10 m) |
| 2 (Green) | 0.530–0.590 μm (30 m) | 0.543–0.578 μm (10 m) |
| 3 (Red) | 0.640–0.670 μm (30 m) | 0.650–0.680 μm (10 m) |
| 4 (NIR) | 0.850–0.880 μm (30 m) | 0.785–0.900 μm (10 m) |
| 5 (SWIR−1) | 1.570–1.650 μm (30 m) | 1.565–1.655 μm (20 m) |
| 6 (SWIR−2) | 2.110–2.290 μm (30 m) | 2.100–2.280 μm (20 m) |

## 3. Normalization Algorithms

We used an approach of hyperspectral library−based normalization. As satellite sensors differ, the influence of sensor channels is best analyzed using radiometric normalization from hyperspectral library data. Polygon features (pseudo−invariant polygon features−based normalization) and point feature normalization (pseudo−invariant point features−based normalization) were compared to determine the most effective method. To identify the effects of adjacent bands, multiple−band fitting and single−band fitting

methods were applied as cross−integrations. Therefore, pseudo−invariant point features with the single band (point−single), pseudo−invariant point features with multiple bands (point−multi), hyperspectral features with the single band (SpecLib−single), hyperspectral features with multiple bands (SpecLib−multi), pseudo−invariant polygon features with a single band (polygon−single), and pseudo−invariant polygon features with multiple bands (polygon−multi) were all tested to evaluate data normalization. Histogram matching was another approach we used for comparison.

The main processing procedures we used were: (1) download valid and contemporaneous images of Sentinel−2A and Landsat 8−OLI and resample Sentinel−2A images to have the same spatial resolution as Landsat 8−OLI (30 m per pixel) using nearest neighbor interpolation; (2) perform spatial registration through geographical coordinates and manual processing; (3) select feature points in each ground type based on ground survey information; (4) confirm the best processing window size and calculate the mean reflectance of each polygon feature area; and (5) calculate fitting equations using the polygon features and couple them with single−band and multiple−band regression for image radiometric normalization.

We examined two factors, correlation coefficient ($R^2$) and root mean square error (RMSE), to determine the accuracy of both regression equations and normalization results. They are briefly described in Equations (1) and (2):

$$R^2 = \frac{Cov(S, \hat{S})}{\sqrt{Var(S) \cdot Var(\hat{S})}} \tag{1}$$

$$RMSE = \sqrt{\frac{1}{N} \sum_{i=1}^{N} (S - \hat{S})^2} \tag{2}$$

where $Cov(X, Y)$ is the covariance of X and Y, and $Var(X)$ is the variance of X. When the two factors were applied to evaluate linear regression, $S$ and $\hat{S}$ are the measured and predicted value, respectively. These represent the reference and normalized subject image reflectance, respectively, when used for normalization evaluation. $N$ is the image pixel number.

### 3.1. Hyperspectral Library−Based Normalization

The key step to normalizing spectral radiation from different sensors is developing a hyperspectral library. We built a hyperspectral library by selecting images of relevant land cover ground types from the USGS and ASTER libraries, and simulated spectra from the PROSAIL model, spectra of GF−5, and Hyperion imagery. Each band's spectral response function from Sentinel−2A and Landsat−8 OLI images (Figure 2) was used in Equation (3) by using corresponding data types from all bands within the same range to obtain the normalized reflectance.

$$\rho_\lambda = \int_{\lambda_1}^{\lambda_2} f_i \rho_i d\lambda_i \tag{3}$$

Where $\rho_\lambda$ is the normalized reflectance on the band (Table 1), $\lambda_1$ and $\lambda_2$ are the start–stop wavelengths of the band $\lambda$, and $f_i$ and $\rho_i$ are the spectral response function and hyperspectral reflectance of band I, respectively. After applying Equation (3) to process each spectral dataset, differences among sensor bandwidths were removed and the corrected data regressed to calculate the transfer coefficient for each band between the two datasets.

We found that radiation from two images has a strong linear relationship with a fitted correlation coefficient approaching 1 (Figure 3). Adjacent bands have little influence on satellite image band radiation (Table 2). Except for band 4, all other coefficients of corresponding bands are close to 1. It is clear from these relationships that the radiometric difference caused by bandwidth effects can be linearly calibrated. If linear regression is used for the radiometric normalization of two images, the bandwidth variation detected is the linear regression coefficient, which is not necessary to conduct separated processing.

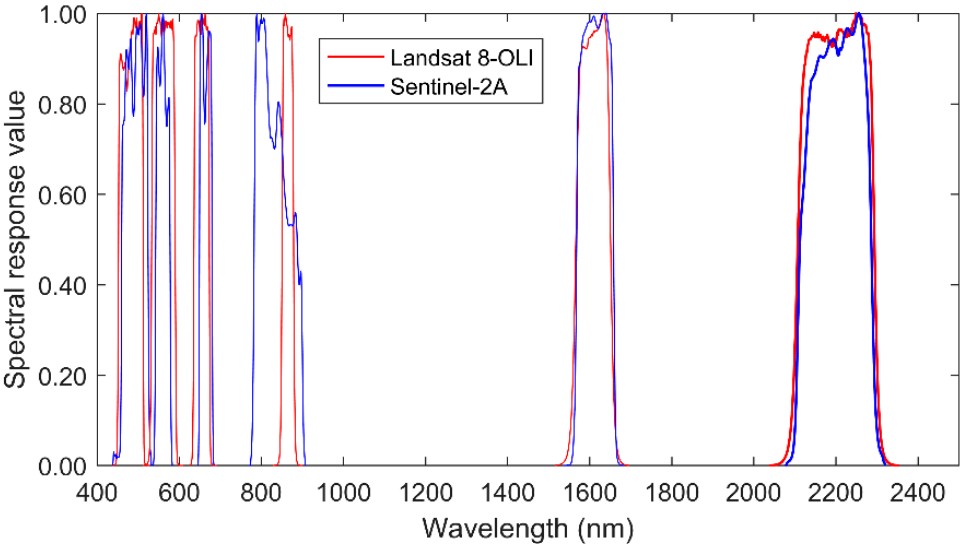

**Figure 2.** The spectral response of Sentinel−2A and Landsat−8 OLI.

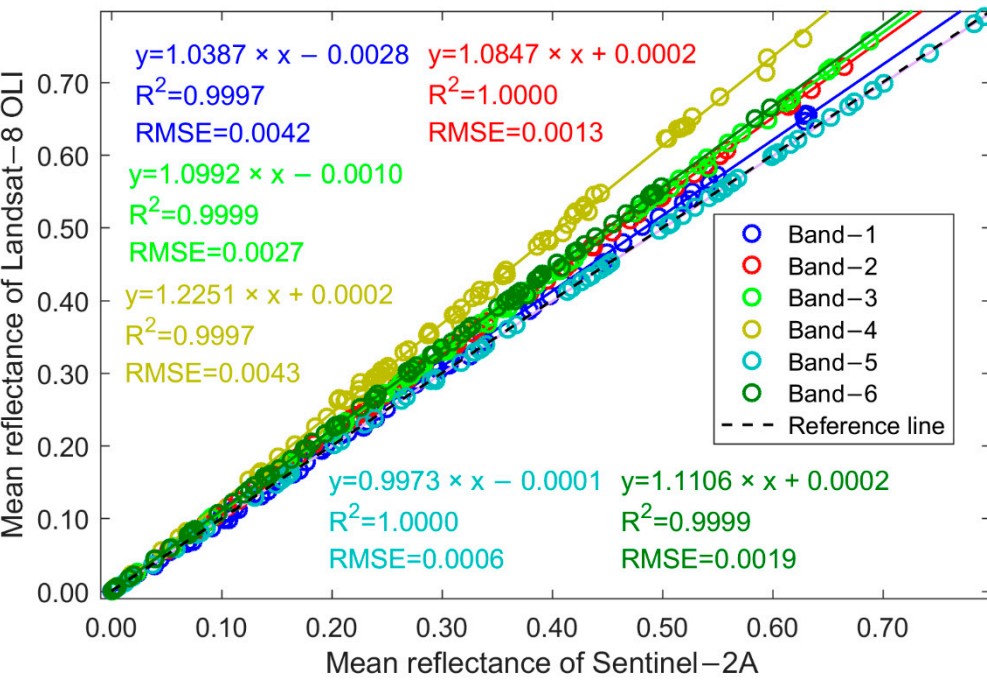

**Figure 3.** Normalization coefficient and errors by SpecLib−Single.

**Table 2.** Normalization coefficient and errors by SpecLib−Multi.

| Sentinel−2A \ Landsat8 | Band−1 | Band−2 | Band−3 | Band−4 | Band−5 | Band−6 |
|---|---|---|---|---|---|---|
| Band−1 | 1.1105 | 0.0329 | −0.0052 | 0.1199 | −0.0023 | 0.0091 |
| Band−2 | −0.0871 | 0.9900 | 0.0523 | −0.1169 | 0.0089 | −0.0262 |
| Band−3 | 0.0626 | 0.0802 | 1.1079 | −0.1476 | −0.0111 | 0.0234 |
| Band−4 | −0.0614 | −0.0279 | −0.0660 | 1.4037 | 0.0049 | −0.0104 |
| Band−5 | −0.0093 | −0.0020 | 0.0000 | 0.0038 | 0.9994 | 0.0020 |
| Band−6 | 0.0095 | 0.0025 | 0.0020 | 0.0031 | −0.0021 | 1.1088 |
| Bias | 0.0002 | 0.0001 | 0.0002 | −0.0001 | 0.0000 | 0.0002 |
| R2 | 1.0000 | 1.0000 | 1.0000 | 0.9999 | 1.0000 | 0.9999 |
| RMSE | 0.0015 | 0.0006 | 0.0006 | 0.0024 | 0.0005 | 0.0019 |

### 3.2. Pseudo-Invariant Point Feature-Based Normalization

Pseudo-invariant polygon feature-based normalization was conducted using pseudo−invariant point features. One crucial procedure for point feature−based radiometric normalization is to accurately select pixels that indicate the same geometric position in different images [22]. In our study, eight different surface types with diverse reflectance were selected to generate the regression equation. To avoid point number weight effects, we chose the same number of samples for each surface class in this process of radiometric normalization based on pseudo-invariant point features.

#### 3.2.1. Pseudo-Invariant Point Features with the Single Band (Point-Single)

The invariant features with single−band regression is:

$$Y_i = aX_i + b \tag{4}$$

where $X_i$ and $Y_i$ are the $i$th band of subject and reference image data. $a$ and $b$ are the normalization coefficient and bias, respectively.

Fifty spectra from each surface type in Landsat-8 OLI images (reference) and Senitnel-2A images were selected. Band data were used to calculate the least square fitting equation and calibrate the radiation data from Sentinel-2A at the reference image's scale (Figure 4). These analyses indicate that the fitting coefficients are less than 1, meaning that all band reflectance from Sentinel-2A are higher than the reference of Landsat-8 OLI data. Among the spectra from eight surface types, all except for man-made, artificial surfaces are clustered. The complexity of artificial pixels, their geometrical location, and spatial resolution increases the difficulty of accurately matching these point features from two different sensor images.

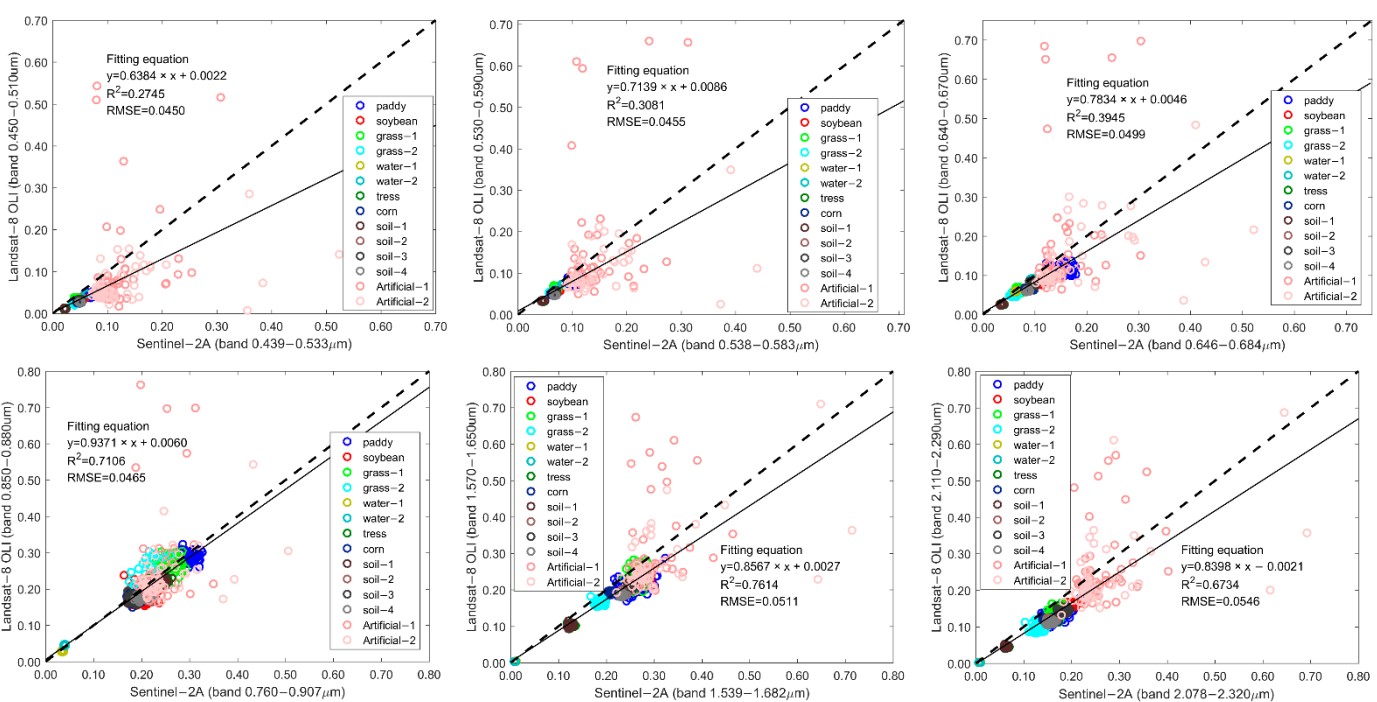

**Figure 4.** Statistical results by pseudo−invariant point feature with single−band data.

#### 3.2.2. Pseudo−Invariant Point Features with Multiple Bands (Point−Multi)

To account for the effects of adjacent bands, all bands in the reference images were compared using multivariate regression. The equation for invariant features with multiple−band regression is expressed as:

$$Y_i = \sum_{i=1}^{n} a_i X_i + b \tag{5}$$

where $a_i$ is the $i$th band normalization coefficient and $n$ is the band number of the subject image.

The coefficient and fitting error were estimated from invariant point features using multivariate regression (Table 3). $R^2$ and RMSE show that the fitting accuracy of the red, green, and blue bands is not the same as for bands 4–6. This appears to be because the first three bands have low reflectance in both Landsat−8 OLI and Sentinel−2A images. Fit using multiple bands was consistently more accurate than fit using single bands, but with low fluctuation (Figure 3). Except for band 4, all bands are significantly affected by adjacent bands, some with negative influences.

**Table 3.** Transfer coefficients and error of point−multi between Sentinel−2A and Landsat−8 OLI.

| Landsat8 <br> Sentinel−2A | Band−1 | Band−2 | Band−3 | Band−4 | Band−5 | Band−6 |
|---|---|---|---|---|---|---|
| Band−1 | 0.6386 | 0.5386 | 0.5773 | −0.0735 | 0.5868 | 1.0106 |
| Band−2 | −0.5458 | −0.3355 | −0.8125 | 0.0166 | −1.2052 | −1.4804 |
| Band−3 | 0.1608 | 0.1515 | 0.7489 | −0.1694 | 0.0999 | 0.1242 |
| Band−4 | 0.0768 | 0.1005 | 0.0639 | 0.9756 | 0.1713 | 0.1599 |
| Band−5 | −0.5442 | −0.5553 | −0.5363 | −0.3156 | 0.1266 | −0.6149 |
| Band−6 | 0.6779 | 0.6665 | 0.6427 | 0.4385 | 0.8774 | 1.5224 |
| Bias | 0.0286 | 0.0443 | 0.0432 | 0.0182 | 0.0440 | 0.0412 |
| $R^2$ | 0.2982 | 0.3356 | 0.4222 | 0.7169 | 0.7733 | 0.6855 |
| RMSE | 0.0441 | 0.0445 | 0.0486 | 0.0460 | 0.0497 | 0.0535 |

### 3.3. Pseudo−Invariant Polygon Features−Based Normalization

Due to spatial resolution differences, the true geometrical position of each pixel covers a unique range. This may result in very different data for the same locations from different sensors. To mitigate this effect, we expanded pseudo−invariant features from polygon points using the mean reflectance of selected areas to replace reflectance from selected pixels. This has the effect of lowering the geometrical error, thus, reducing the uncertainty that may be caused by non−corresponding points.

#### 3.3.1. Pseudo−Invariant Polygon Features with the Single Band (Polygon−Single)

To avoid the influence of noncorresponding scatter points, the mean reflectance of selected areas was substituted for pixel reflectance in regression analyses. When fitting a single band, a much more precise equation is generated than that obtained from point features. Correlation coefficients are all greater than 0.9, except for band 3 (Figure 5).

#### 3.3.2. Pseudo−Invariant Polygon Features with Multiple Bands (Polygon−Multi)

Multivariate regression equations have similar high accuracy as single−band regression, even obtaining more precise correlations (Table 4). Bands 4, 5, and 6 are strongly and positively influenced by corresponding bands while the others are greatly affected by adjacent bands.

**Table 4.** Transfer coefficient and error of polygon−multi between Sentinel−2A and Landsat8−OLI.

| Landsat8 <br> Sentinel−2A | Band−1 | Band−2 | Band−3 | Band−4 | Band−5 | Band−6 |
|---|---|---|---|---|---|---|
| Band−1 | 0.6945 | −0.0268 | −0.0758 | 0.2168 | −0.1613 | 0.1875 |
| Band−2 | 0.2421 | 1.2244 | 0.7760 | 0.1973 | 1.0132 | 0.5372 |
| Band−3 | −0.1027 | −0.1630 | 0.4461 | −0.0332 | −0.4453 | −0.3343 |
| Band−4 | −0.0455 | −0.0540 | −0.0878 | 0.9290 | 0.0004 | 0.0350 |
| Band−5 | 0.1315 | 0.1467 | 0.1451 | 0.2432 | 0.8925 | −0.0506 |
| Band−6 | −0.1596 | −0.1671 | −0.1581 | −0.3065 | 0.0135 | 0.8808 |
| Bias | −0.0070 | −0.0113 | −0.0123 | −0.0240 | −0.0368 | −0.0293 |
| R2 | 0.9667 | 0.9743 | 0.9785 | 0.9956 | 0.9977 | 0.9972 |
| RMSE | 0.0020 | 0.0021 | 0.0028 | 0.0043 | 0.0035 | 0.0029 |

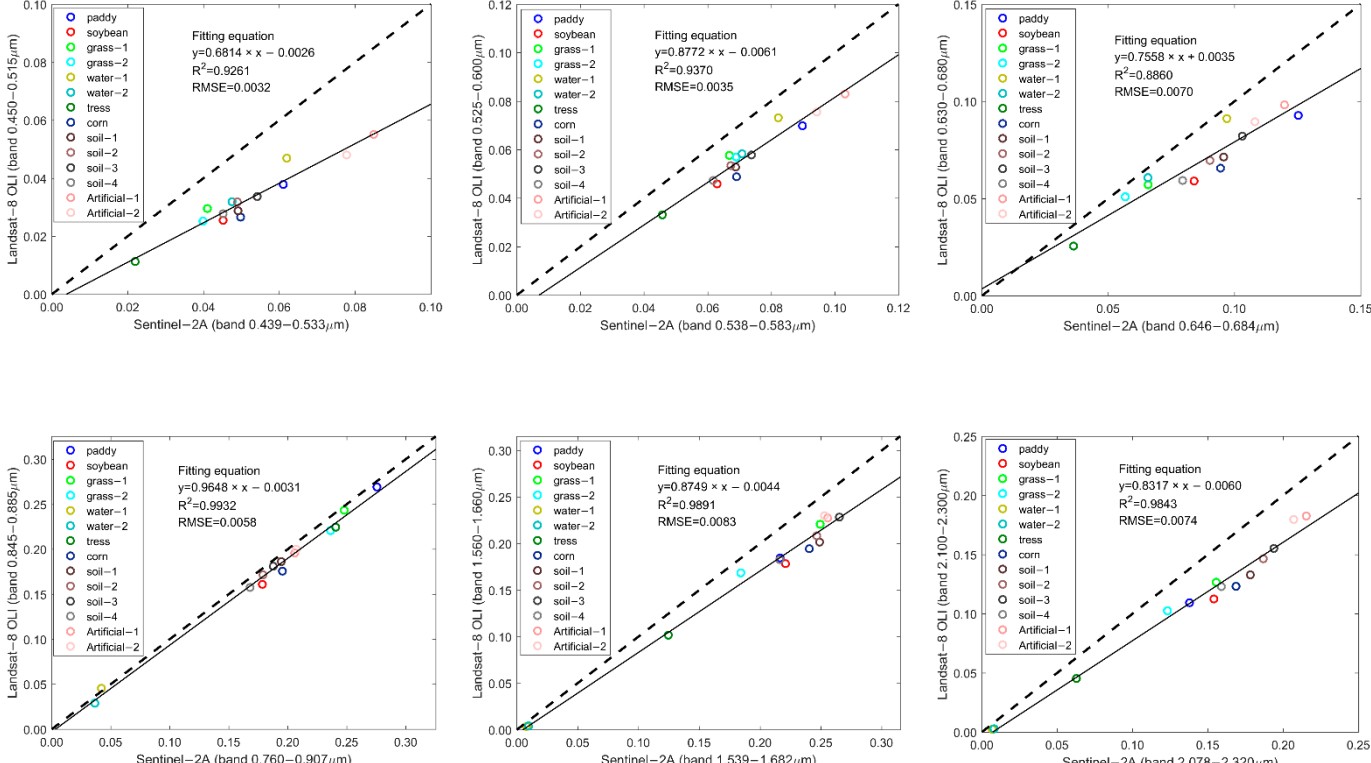

**Figure 5.** Statistical results by pseudo−invariant polygon features with single−band data.

### *3.4. Histogram Matching*

Histogram matching is one of the most common distribution−based radiation normalization methods [35]. It can avoid subjectivity in the selection of pseudo point features as well as image misregistration [36]. One simple way to conduct histogram matching is to plot the histogram of the reference and subject images and use the mean differences between the two to shift, or normalize, the subject histogram to the master [7]. The principle of histogram matching is to calculate the frequency of image brightness accounting for frequency variation. This approach has the clear advantage of using all pixels from images. Here, reference and subject images are manually geospatially calibrated, and histogram matching is applied to obtain a normalized comparison object.

## 4. Evaluation of Normalization Results

The Sentinel−2A data were calibrated using regression equations (above) and then compared to the reference image. We use three perspectives to represent the accuracy of calibrated results: band comparison, pixel comparison, and reflectance frequency.

### *4.1. Comparisons among Bands*

To compare evaluation methods among bands, scatter analysis was applied to the normalized subjects and reference images. The coefficient and bias of fitting equation results are shown in Figure 6; normalization is much more precise when the fitting coefficient is closer to 1 and the bias is closer to 0. Except for band 2 of point−multi, coefficients obtained from polygon−multi are much closer to 0, which means the reflectance of the two images has better radiometric consistency. The polygon−single method follows closely, and then, in order, SpecLib−single and SpecLib−multi that only consider bandwidth. The fitting coefficients are less than 1, which is counter to results from other methods. The bias results show that the polygon−single and polygon−multi methods are relatively more accurate for most bands. As indicated by $R^2$ and RMSE evaluation (Figure 7), polygon−multi has the best fit and point−multi the worst, indicating that the calibrated subject image has satisfactory radiometric consistency with the reference image.

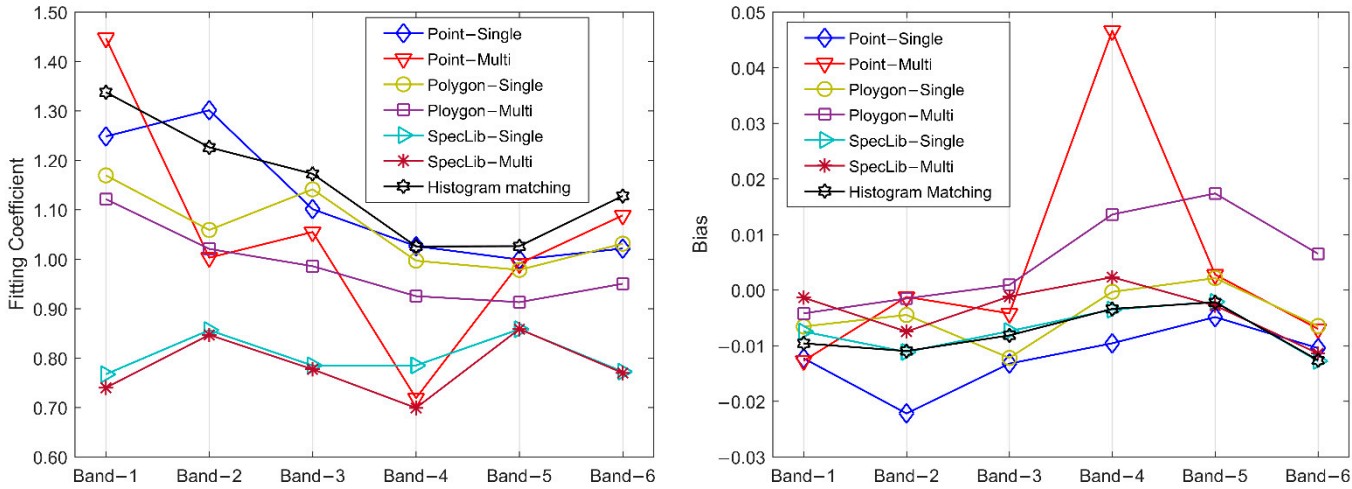

**Figure 6.** Scatter−fitting coefficient and bias between reference and subject images.

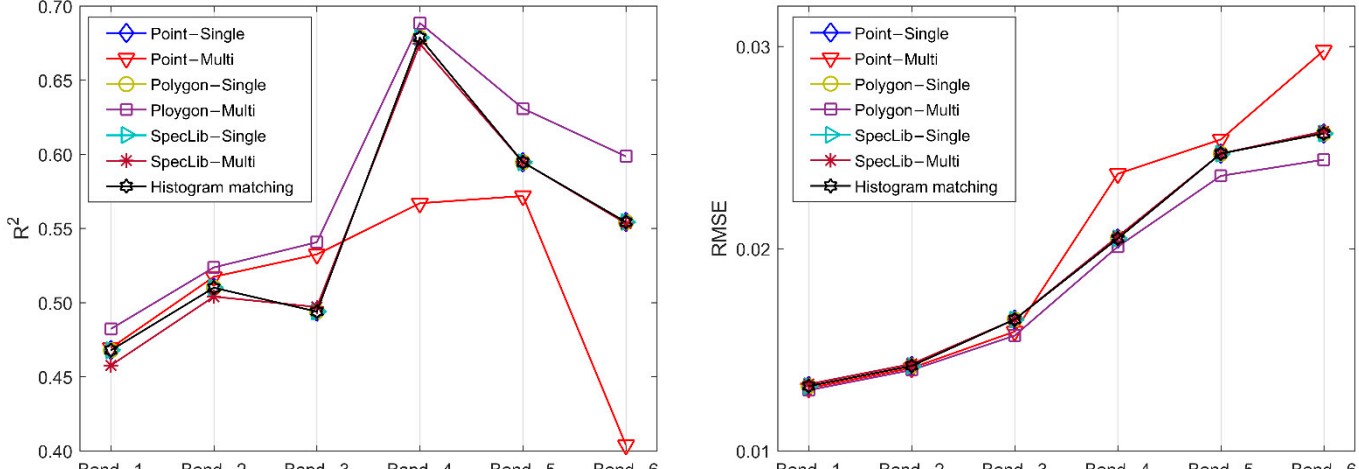

**Figure 7.** Evaluation of scatter−fitting results.

### 4.2. Comparison of Each Pixel Normalization

The ultimate purpose of radiometric normalization is to obtain spectral curve consistency at each pixel. In both $R^2$ and RMSE maps, hyperspectral library−based normalization methods have the lowest $R^2$ and the highest RMSE (Figure 8). Among the other methods, point−single and histogram matching have similar accuracy compared to polygon−single and polygon−multi. Except for pixels in the water spectrum (blue and dark blue in the $R^2$ histogram matching map) and artificial (brightly colored areas in the RMSE point−multi map), the pseudo−invariant polygon features−based normalization has significantly greater accuracy than other methods. Pseudo−invariant polygon features−based normalization has larger $R^2$ and smaller RMSE (Table 5). As the study area is an agricultural planting region, artificial pixels are few, slightly improving the accuracy of polygon−based normalization.

### 4.3. Comparison of Image Frequency Distribution

After determining single and multivariate regression to the subject image, the reflectance frequency of each band was generated to compare with the frequency from the reference image (Figure 9). The more the frequency curves overlap, the better the radiometric normalization result. The subject image frequency curves from all methods, except for hyperspectral library−based normalization, are nearly identical to those of the reference image reflectance frequency in bands 4, 5, and 6. Curve peaks are much closer between

two images in all bands compared with the original frequency. Polygon−multi, histogram matching, point−multi, and polygon−single perform better, matching the frequency peak position. However, within bands 2 and 3, the frequency peak of subject images from point−multi and polygon−single appear to be offset from the frequency peaks of reference images. Point−multi, polygon−single, and polygon−multi methods provide the nearest matching frequency peaks between two images. In these two areas, polygon−multi has the highest radiometric normalization accuracy.

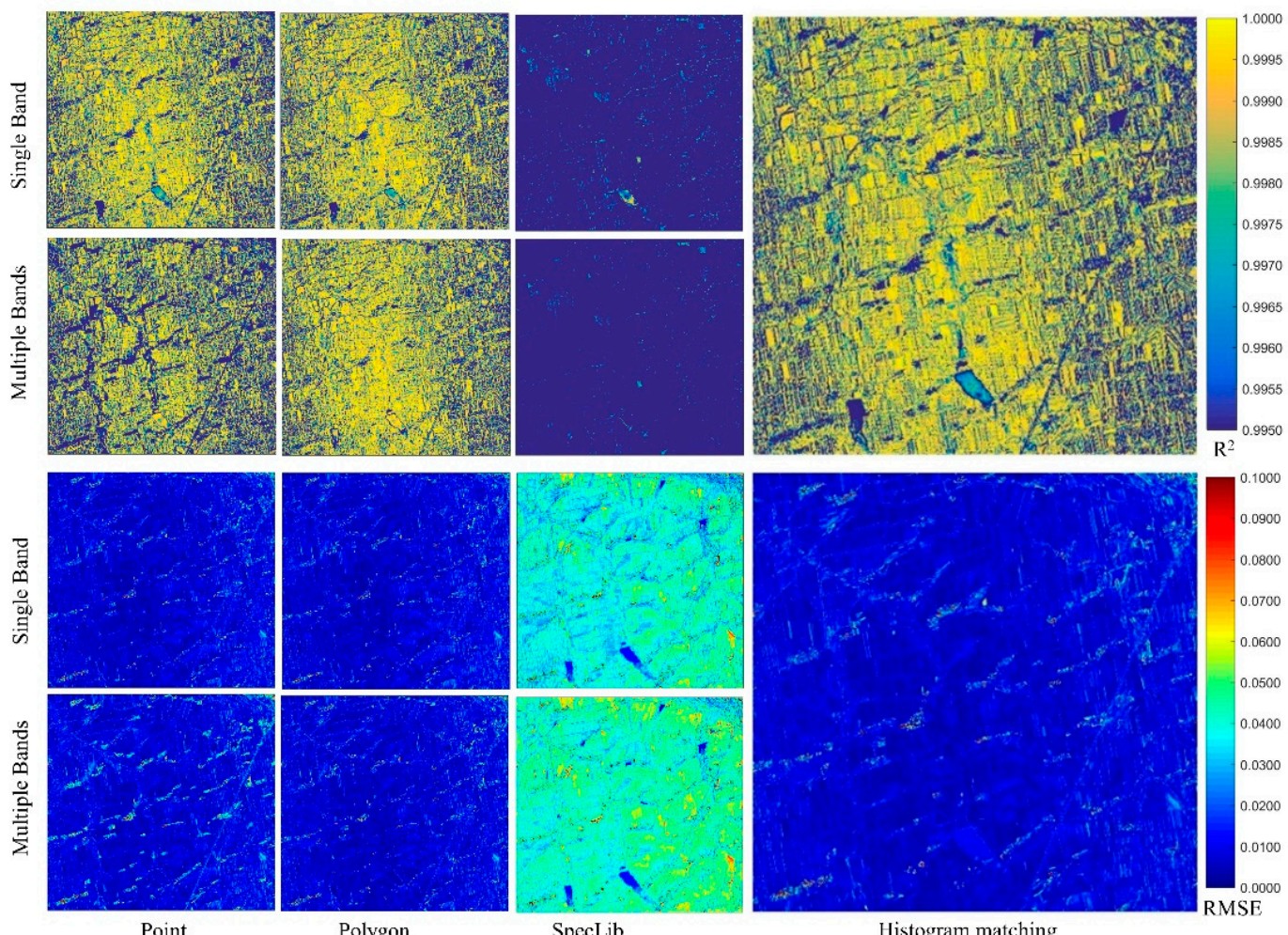

**Figure 8.** $R^2$ and RMSE of image consistency at each pixel. Note, to present the pixel−wise values, the minimum and maximum of R2 maps range from 0.9950 to 1, and RMSE maps range from 0 to 0.1.

**Table 5.** The mean $R^2$ and RMSE of image consistency at each pixel. To quantitatively present accuracy, we computed the mean value of each map.

| Error \ Methods | Point−Single | Point−Multi | Polygon−Single | Polygon−Multi | SpecLib−Single | SpecLib−Multi | Histogram Matching |
|---|---|---|---|---|---|---|---|
| $R^2$ | 0.9944 | 0.9941 | 0.9945 | 0.9948 | 0.9861 | 0.9812 | 0.9943 |
| RMSE | 0.0112 | 0.0148 | 0.0097 | 0.0095 | 0.0398 | 0.0439 | 0.0095 |

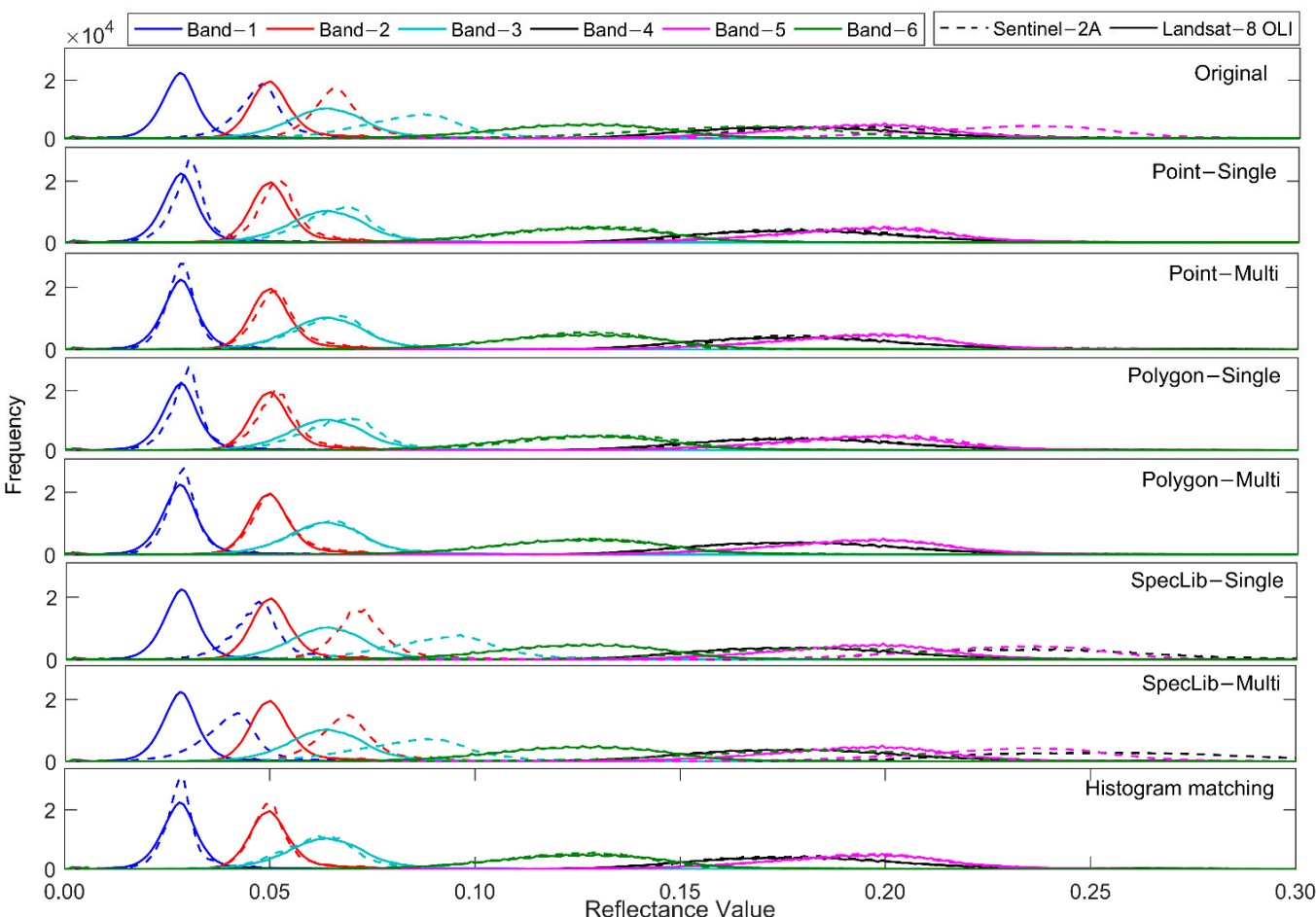

**Figure 9.** Reflectance frequency comparison of calibrated and reference images.

## 5. Discussion

### 5.1. Variation of Pseudo−Invariant Point Features

The fitting coefficient of the point−single (Figure 4) and point−multi (Table 3) methods are close to their respective mean reflectance variation (Figure 5 and Table 4) at each band. Their correspondence indirectly explains the stability of regression relationships obtained for polygon features. Indeed, it is artificial surfaces (Figure 4) that appear to vary irregularly, inducing variation in regression correlations, and resulting in inaccurate normalization results. We selected non−adjacent artificial points in the study area and used them to center the chosen polygon region, point variation, and mean value of 115 different rectangle regions (each rectangle includes 5 × 5 pixels; Figures 10 and 11). All bands have pixels greater than 0.5 and lower than 0.1 in reflectance, and the higher and lower reflectance pixels are not located in the same place (pixel No. in $x$−axis). If those pixels are considered as pairwise points, a regression analysis may vary and not be universal. Using the mean reflectance of a 5 × 5 pixel area can reduce the influence of bright or dim pixels, increasing the similarity of artificial pixel reflectance. By increasing the polygon size from 3 × 3 to 101 × 101 pixels, the variance of mean reflectance in each band is affected (Figure 12). The lower the variance is, the fewer abnormal artificial points the image contains, and with the increase in polygon size, the variance of mean artificial reflectance decreases. In addition, we use all the image pixels in linear regression (Figure 13), compared with the polygon−single (Figure 5), and found they were significantly affected by the artificial points. When the polygon size is large, the proportion of abnormal pixels decreases, but the area will contain more land cover types and weaken the effect of artificial features. To keep artificial spectral features, setting a suitable polygon size is essential for automatically selecting polygon features for radiometric normalization.

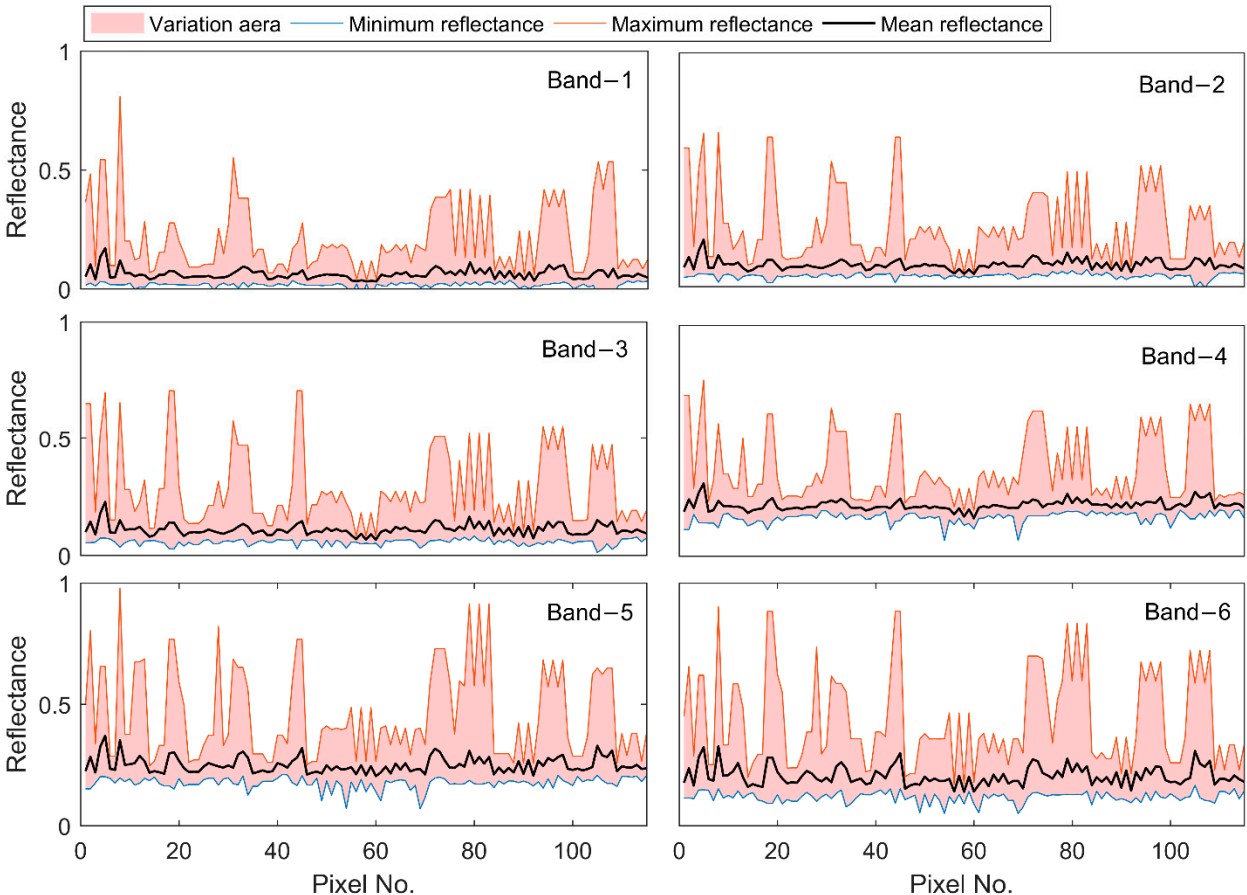

**Figure 10.** Reflectance variation of 5 × 5 artificial pixels from Landsat−8 OLI.

### 5.2. Limitations of Polygon Feature−Based Normalization Algorithms

For the most widely used relative radiometric normalization algorithms, automatically selecting pseudo−invariant point features is the most important step [37]. How to extract pseudo−invariant polygon features effectively and precisely is, therefore, fundamental to preprocessing. In Section 4.1, we discussed the influence of polygon size. Integrating the pseudo−invariant points with a suitable polygon size, as demonstrated by Kim, Pyeon [38], Zhou, Liu [24], and Lin, Wang [25], is a practical way to automatically select polygon features and will be included in future research and monitoring.

In this study, we only considered contemporary remote sensing images. Due to changing atmospheric conditions over time, regression relationships are not universal, especially in temporal sequences. When normalizing temporal image radiation, it is necessary to extract multiple invariant polygon characteristics and establish separate regression equations for each. Over the history of remote image data collection, most differences in radiation have been caused by atmospheric conditions. This kind of radiation normalization usually relies on roads, developments, and other types of artificial surfaces for georeferencing. However, different image resolutions and spatial mismatches may result in the same pixel from two images containing different landcover types and different endmember abundance. Unlike artificial surfaces, land cover types usually change with the seasons, thus, image data across periods may not maintain radiation consistency at the pure pixel scale. However, mixed pixel decomposition can extract pure artificial surface radiation of invariant feature points and improve radiation normalization accuracy. Our study highlights avenues to be explored in future research relying on polygon feature−based normalization. We considered a variety of ground object types in a ground survey. PIFs were determined by manual processing, which will affect the application in an area with or without limited samples. Classification methods to automatically extract PIFs are active

methods that will be discussed in future studies, as will the comparison to some classical algorithms, such as IRMAD [28] and key−points−based RRN [18,39], to identify the accuracy and efficiency of these algorithms.

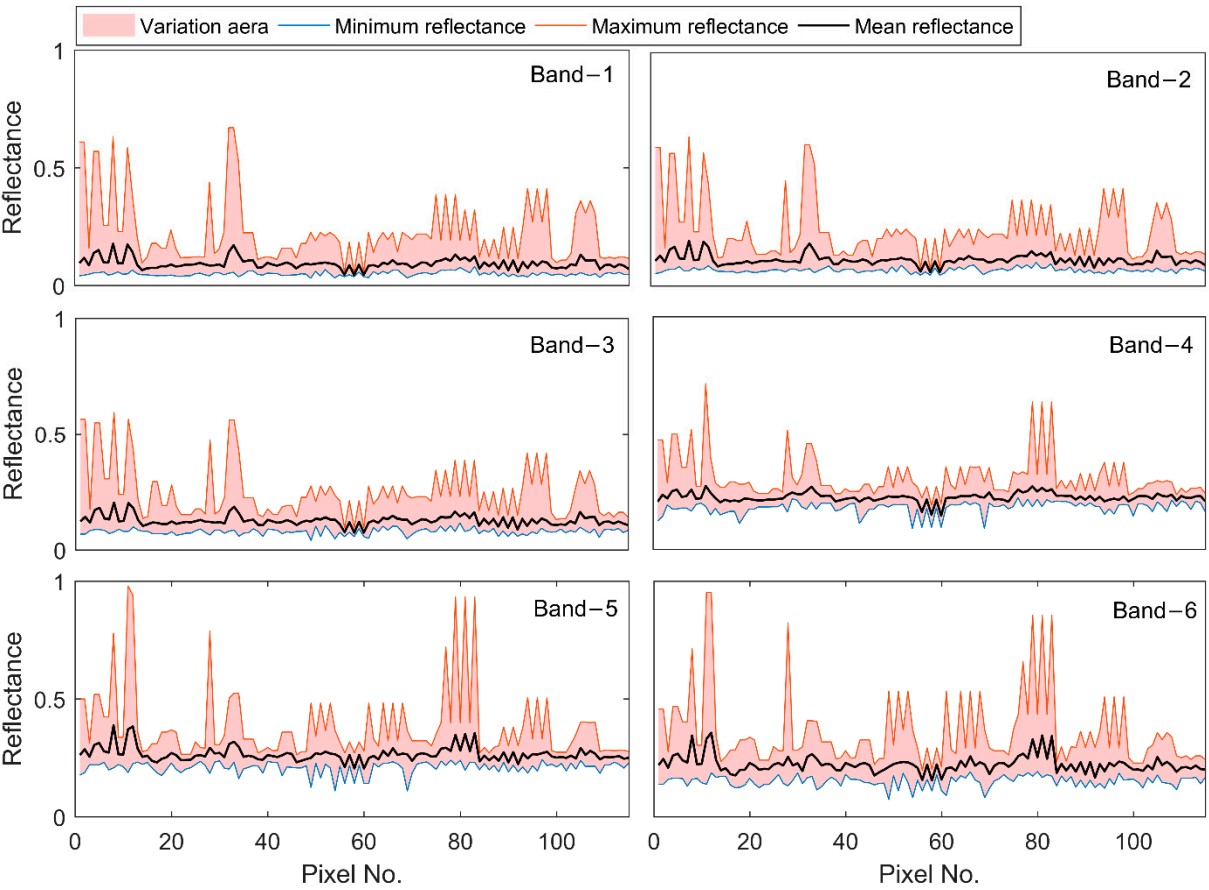

**Figure 11.** Reflectance variation of 5 × 5 artificial pixels from Sentinel−2A.

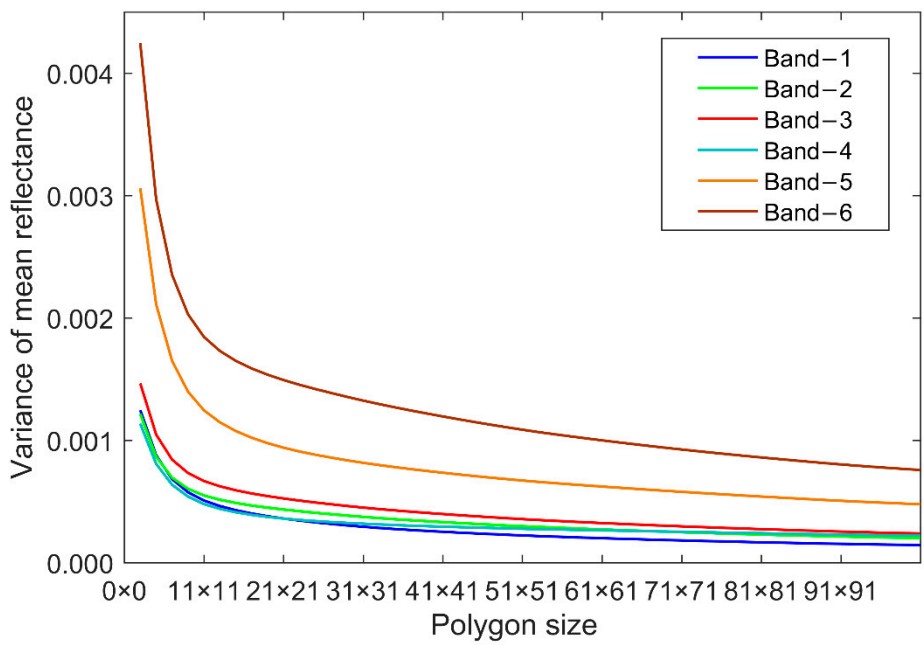

**Figure 12.** Variance analysis of mean reflectance among artificial polygon sizes.

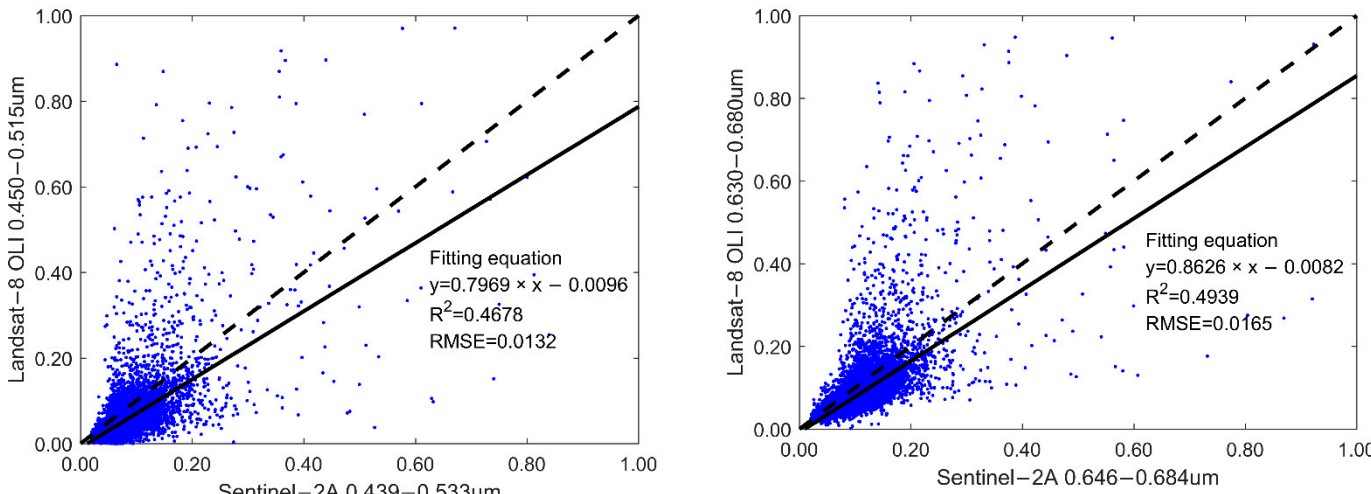

**Figure 13.** Using all pixels of two images to fit scatter plots.

## 6. Conclusions

We propose novel relative radiometric normalization algorithms based on pseudo−invariant polygon features with single−band and multiple−band regression, called polygon−single and polygon−multi methods. In comparison to normalization algorithms based on pseudo−invariant point features, hyperspectral features, and histogram matching, the proposed methods are more accurate when using contemporaneous Sentinel−2A and Landsat−8 OLI image radiometric normalization. They are followed by histogram matching, pseudo−invariant point features−based normalization, and hyperspectral library−based normalization. The comparative results in each band show that polygon−multi has the best scatter fitting, especially for bands 2 and 3. Polygon−multi and polygon−single are the most accurate ($R^2$ = 0.9948, 0.9945 and RMSE = 0.0095, 0.0097, respectively) for each pixel comparison. The reflectance frequency of the whole image also illustrates that polygon−multi and polygon−single have the best shape fitting and are close to the histogram matching algorithm. Our multiple−band regression analysis indicates that the adjacent bands have a large influence and the fit of pseudo−invariant features using multiple−band data achieved is higher and provides more accurate normalized reflectance than that with a single band. Using the mean reflectance of selected polygons to replace point reflectance can effectively eliminate the influence of abnormal (artificial) points, making the regression equation more universal. The proposed algorithms have distinct advantages over other methods, especially for the normalization of artificial surface pixels.

**Author Contributions:** Methodology, L.C., Y.M. and Y.Y.; Validation, Y.L. (Yi Lian) and Y.L. (Yanzhen Lin); Investigation, H.Z. and Y.L. (Yanzhen Lin); Writing—original draft, L.C. All authors have read and agreed to the published version of the manuscript.

**Funding:** This research was funded by the Project of Special Investigation on Basic Resources of Science and Technology grant number 2019FY202501 and National Natural Science Foundation of China grant number 41802246 and 41971306.

**Institutional Review Board Statement:** Not applicable.

**Informed Consent Statement:** Not applicable.

**Data Availability Statement:** The datasets that were analyzed in this study can be found publicly here: https://scihub.copernicus.eu/ (accessed on 26 September 2019) and https://earthexplorer. usgs.gov/ (accessed on 22 September 2019). The dataset presented in this study can be available on request from the author.

**Acknowledgments:** The authors would like to thank the United States Geological Survey and European Space Agency for freely providing remote sensing images.

**Conflicts of Interest:** The authors declare no conflict of interest.

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
