# Peer review of "Radiometric Normalization Using a Pseudo−Invariant Polygon Features−Based Algorithm with Contemporaneous Sentinel−2A and Landsat−8 OLI Imagery"

_applsci, doi:10.3390/app13042525_

Round 1

Reviewer 1 Report

This study presents the relative radiometric normalization method to improve multi-image consistency with existing information, and also propose Landsat-8 OLI as the reference image and Sentinel-2A as the subject image, to apply pseudo polygon features-based algorithms with polygon features through the single band and multiple band regression.  This is an interesting study and deals with some real issues of remote sensing filed. The paper is well-founded and can be publish after incorporating the suggested comments.

1.    In abstract section, the discussion on findings is limited and remove the unnecessary details of your methodology and discuss your findings to create the interest of readers.

2.    The introduction section is sound. But I think the authors still needs to provide a sound justification on why this current research is necessary beyond what previous studies reported by clearly specifying the aim and innovation.

3.    To strengthen the literature review, latest studies (2020-2021) must be discussed and cited in this study

4.    Clearly indicate the benefits of your proposed radiometric normalization method in introduction part, and also discuss what is your contribution to develop this method.

5.    Visibility, formatting and resolution of all Figures are not match the requirements of journal. Prepare these figures according to publisher guidelines and standards.

6.    Conclusion section only contains the scope of this study. Discuss the findings of your study in details with evidence and arguments

Reviewer 2 Report

The authors presented a method for relative radiometric calibration of Landsat 8 images based on the sentinel 2 images. As for me, relative radiometric normalization is an interesting topic and the paper can be improved if the authors consider my comments. I have some concerns and suggestions that the authors should be applied before the manuscript is considered for publication in the journal. 

1- The introduction needs to improve with a strong literature review. In recent years, Relative radiometric normalization was improved by many authors and the authors should consider new methods for literature review. I suggest using the following citations to improve the literature review. 

https://doi.org/10.3390/rs9121319

https://doi.org/10.3390/rs14081777

https://doi.org/10.1080/01431161.2021.1934912

https://doi.org/10.1109/TGRS.2020.2995394

https://doi.org/10.1109/LGRS.2020.3031398

https://doi.org/10.1109/LGRS.2020.3047344

https://doi.org/10.1080/01431161.2016.1213922

https://doi.org/10.1016/j.rse.2007.07.013

https://doi.org/10.1109/JSTARS.2020.2971857

https://doi.org/10.3390/rs13163125

2- We have two kinds of methods for PIF-based relative radiometric methods, which are mainly divided into two categories: non-PIF-based and PIF-based methods [I][II]. For example, non-PIF-based uses the whole of the images for radiometric adjustment, while the PIF-Based methods use time-invariant features. please academically bring these sentences by using the following citations. 

[I] https://doi.org/10.1016/j.apm.2013.01.006

[II] https://doi.org/10.1109/JSTARS.2021.3069919

3- Page 2, Line 73-75: "During both manual and automatic feature 73 selection, there are two issues: the spectra of paired pixels may differ due to the pixel-level spatial mismatch, and a regression analysis with one variable does not consider the influence of adjacent bands." This issue is well addressed by the following citations for registered and non-registered bitemporal images acquired by the inter/intra sensors. 

[A] https://doi.org/10.1109/JSTARS.2021.3069919

[B] https://doi.org/10.1109/TGRS.2021.3063151

[C] https://doi.org/10.1080/01431161.2022.2102951

[D] https://doi.org/10.3390/rs13193990

However, the authors can use this sentence instead of the first issue:

Although spatial mismatch problems in RRN are well addressed in the keypoint-based methods [A][B][C][D], they are computentially intensive and not appropriate for the images with a few misregistration errors (e.g., Landsat 8 and Sentinel 2). 

4- In the last paragraph of the introduction, please state the novelties of the present study in summary.  

5- To well validate the presented method, the authors should apply the method on more than one image pair/dataset which is acquired from a different area. Therefore, please apply the method to other datasets and study areas. 

6- The method should be compared with another well-known method like IRMAD [x] or keypoint-based RRN [xx]. The codes of both methods are available on GitHub and the internet. The computation time should be reported, as well. 

[x] https://doi.org/10.1016/j.rse.2007.07.013

[xx] https://doi.org/10.1109/JSTARS.2021.3069919

7- In Table 1 . please refer to the name of the band not the number of the bands. This is because, for example, the number of NIR and SWIR bands in the Sentinel 2 and Landsat 8 is different. Please add the radiometric and spatial resolution of considered sensors in different bands. 

Please add a flowchart in the methodology section. I think this is very important for the reader to see the workflow of the presented method, even in a simple way. 

8- The experimental results were well presented and can be completed by the results of another dataset and comparison with a well-known method like IRMAD. Moreover, please present the visual results of the normalized image because this is very important. I am aware you can not present the 16-bit results, but you can use QGIS to present them and get 8-bit output. 

Round 2

Reviewer 2 Report

The authors well adressed all of my comments.